# Utility of Three-Dimensional Printed Model in Biventricular Repair of Complex Congenital Cardiac Defects: Case Report and Review of Literature

**DOI:** 10.3390/children9020184

**Published:** 2022-02-01

**Authors:** Lauren Gabriel Betancourt, Si Hui Wong, Harinder R. Singh, Daniel Nento, Arpit Agarwal

**Affiliations:** Children’s Hospital of San Antonio, San Antonio, TX 78207, USA; laurenashleygabriel@gmail.com (L.G.B.); si.wong@christushealth.org (S.H.W.); Harinder.Singh@bcm.edu (H.R.S.); Daniel.Nento@bcm.edu (D.N.)

**Keywords:** congenital heart disease, 3D printing, heterotaxy syndrome, modified Warden procedure

## Abstract

Heterotaxy is a rare syndrome associated with cardiac complexity, anatomic variability and high morbidity and mortality. It is often challenging to visualize and provide an accurate diagnosis of the cardiac anatomy prior to surgery with the use of conventional imaging techniques. We report a unique case demonstrating how the use of three-dimensional (3D) cardiac printed model allowed us to better understand the anatomical complexity and plan a tailored surgical approach for successful biventricular repair in a patient with heterotaxy syndrome.

## 1. Introduction

Heterotaxy syndrome is a rare disorder characterized by abnormal lateralization and isomerism of thoraco-abdominal organs across the left–right axis of the body [1,2,3,4]. It is often associated with complex cardiac anatomy, which can include right or left atrial isomerism, ambiguous atrial anatomy, cardiac malposition, common atrioventricular (AV) valve defects, and abnormalities in pulmonary and/or systemic venous connections [5]. The complexity and the severity of the associated defects in heterotaxy syndrome determines the surgical decision for univentricular versus biventricular repair. (Surgical management with univentricular or biventricular repair in patients with heterotaxy syndrome is typically determined by the complexity and severity of the associated cardiac defects) [2]. Given the complex cardiac anatomy associated in heterotaxy syndrome, several tools are used to define the anatomy prior to surgical intervention, including echocardiography, computerized tomography (CT), and Magnetic Resonance Imaging (MRI), but anatomy is often confirmed intraoperatively [2,5]. In order to minimize uncertainty in defining the patient’s anatomy, we utilized 3D-printed cardiac model for pre-surgical planning and familiarize ourselves with the procedures to be performed. To our knowledge, this is the first reported case using 3D printed cardiac model to plan a unique biventricular repair using a Warden-like procedure in a patient with heterotaxy syndrome and complex cardiac anatomy.

## 2. Case Presentation

The patient is a 5-month-old male who was diagnosed with multiple congenital anomalies including heterotaxy syndrome, polysplenia, intestinal malrotation, absent septum pellucidum, cleft lip and palate, small left kidney with hydronephrosis and right inguinal hernia at birth. His post-natal echocardiogram revealed mesocardia with apex to the left, midline liver, atrial situs ambiguous with a common atrium and ipsilateral pulmonary veins (left pulmonary veins draining into the left-sided atrium and right pulmonary veins draining into the right-sided atrium), D-looped ventricles, normally related great arteries (A,D,S), interrupted inferior vena cava (IVC) with hemiazygos continuation to the left superior vena cava (SVC) draining into the left and anterior aspect of the common atrium, hepatic veins draining to the common atrium, a partial AV canal, small anterior muscular ventricular septal defect (VSD), and a moderate sized patent ductus arteriosus (PDA) [6]. Genetic testing detected an interstitial deletion on chromosome 20. The cardiology and cardiovascular surgical team decided to medically manage him with diuretics with a plan for surgical repair at 4 to 6 months of age.

At three and a half months of age, the patient underwent elective cleft lip and palate repair where he experienced acute hypoxemic respiratory failure and cardiorespiratory arrest, as well as multiple failed attempts at extubation. Cardiac computerized tomography angiography (CTA) confirmed the cardiac anatomy previously seen on post-natal echocardiogram (Figure 1 and Figure 2). It also revealed an anomalous origin of an accessory left pulmonary artery (LPA) arising from the proximal right pulmonary artery (RPA), which resulted in severe compression of the left mainstem bronchus against the descending aorta, forming a partial left pulmonary artery sling [6]. Given his complex medical history and cardiac anatomy, a three-dimensional (3D) cardiac printed model was used to further delineate his cardiac anatomy and determine the safest plan for surgical intervention.

The DICOM (Digital Imaging and Communications in Medicine) files from CTA were imported into the segmentation software. Mimics Medical 23 software was used for 3D segmentation. The resulting 3D segmented virtual model was converted to a STL (Stereolithography or Standard Tessellation Language) file that is compatible with 3D printing. The heart model design was then printed using Stratasys J750 printer to generate the flexible cardiac model. Loose powder present at the end of the building process was vacuumed off to reveal the final product. Figure 3 and Figure 4 show the pre-operative 3D cardiac model of this patient. 

Upon examination of the cardiac model, the surgical decision was to perform a complex biventricular surgical repair. At the age of 5 months, the patient underwent partial AV canal repair with closure of mitral valve cleft. A modified Warden procedure was performed. As previously planned on the 3D model, a large systemic venous baffle was created to drain systemic venous circulation into the right atrium. The base of LSVC, left-sided atrial appendage and right-sided atrial appendage were fashioned to form the posterior wall of the baffle. The joint of the LSVC to the left atrium was closed with a pericardial patch. A 30-degree CardioCel patch was used to form the anterior wall of the systemic venous baffle. With that, the LSVC and hemiazygos vein that carries the flow from the IVC was directed into the right atrium. The mitral cleft was closed with an interrupted 6-0 Prolene sutures. To create an atrial septum, a CardioCel bovine pericardial patch was used. The patch was started from the posterior and inferior portion of the left atrium and the suture line was carried out anteriorly. This method allowed us to move toward the space of tissue between the mitral valve and the tricuspid valve. Then, the same patch was turned to the superior portion to create an interatrial septum which was missing in this patient. At the end of the procedure, separation between the right atrium from the left atrium was obtained, incorporating the four pulmonary veins communicating with the mitral valve, and the hepatic veins together with blood coming from the superior vein baffle communicating with the tricuspid valve.

Normal biventricular size and systolic function with no evidence of significant intracardiac shunt was obtained on cardiac MRI performed 5 weeks postoperatively. His postoperative course from the cardiac perspective remained uneventful until discharge [6]. Figure 5 and Figure 6 show the post-operative CT imaging of the patient. Figure 7 compares the preoperative 3D cardiac anatomy and the postoperative 3D cardiac anatomy to demonstrate the baffle connection from left SVC to left-sided atrial appendage and the right-sided atrial appendage.

## 3. Discussion

Historically, heterotaxy patients have had poor surgical outcomes associated with high morbidity and mortality due to structural abnormalities of systemic and pulmonary venous connections, increased incidence of single ventricle physiology, pulmonary and aortic outflow obstruction, arrhythmias and infections associated with splenic dysfunction [1,5,7,8,9,10,11,12,13]. Important principles of cardiac repair in heterotaxy syndrome include division of extracardiac communication, intracardiac separation of venous return between pulmonary and systemic circulation, and the use of baffles and an atrial switch with patches in isomeric cardiac repair [14]. Patients with single ventricle physiology often require multi-step palliation with a Blalock–Taussig shunt and bidirectional Glenn/Kawashima procedure to direct systemic flow to the pulmonary circulation prior to a Fontan procedure, which is considered the definitive intervention [15]. Biventricular repair is considered in heterotaxy patients with adequate biventricular volume, function, and favorable anatomy between AV valves and ventricles [1,5,14]. The surgical techniques described in the literature include intra-atrial and interventricular redirection, AV valve repair and the use of baffles to direct systemic venous return to the right atrium [4,15].

In recent years, 3D printing has been increasingly used in the surgical field, especially for reconstructive surgeries in craniomaxillofacial [16,17], orthopedics [18], and congenital heart diseases to minimize surgical complications. In congenital heart defect surgery, specifically for biventricular repairs, 3D printed cardiac models, in conjunction with other imaging modalities, have helped in pre-surgical planning by defining the relationship of the ventricles, VSD morphology, and the VSD relation to the outflow tracts. This is crucial in determining the feasibility of a biventricular surgical approach [19,20,21,22,23,24,25,26,27]. 3D printed cardiac models have also played a pivotal role in changing the surgical plan from conservative management to biventricular repair, single ventricle palliation to biventricular repair, or modifying the entire surgical approach for biventricular repair [28]. 

In addition, 3D printed heart models have helped reduce communication barriers between patients, their families, and physicians. It aids in the education of cardiac anatomy, pathology, surgical procedures, and post-operative progress [22,29]. These heart models also help strengthen relationships among medical providers as they promote constructive discussions between surgeons and cardiologists [19,30]. Therefore, given the complex anatomy and risks involved in this case, the team decided to utilize 3D printing to: (1) familiarize themselves with the anatomy, evaluate techniques and procedures, (2) formulate a novel surgical approach, and (3) counsel and educate patients and their families pre- and post-operatively.

From extensive literature review, there are 14 reports of successful biventricular repair using 3D printed cardiac models in complex congenital heart disease (Table 1). Two prior reports in the literature demonstrate the use of 3D printed cardiac models in defining complex cardiac anatomy and pre-surgical planning for biventricular repair in patients with heterotaxy syndrome [27,28]. However, neither of these reports described the novel surgical approach used in this case — the “modified Warden procedure”— which involved creating a baffle with the right-sided and left-sided atrial appendages to direct left sided systemic venous return to the right sided atrium.

Agarwal A. discussed the use of 3D cardiac modeling to explore the possibility of biventricular repair in a 4-year-old patient with heterotaxy syndrome as well as its importance in the planning of her complex cardiac surgery [27]. Valverde et al. performed a study to investigate the impact of 3D cardiac models on the surgical planning of patients with complex cardiac anatomy. They reported the positive influence of 3D cardiac model on the management plan of a 34-year-old patient with heterotaxy syndrome [28].

The remaining reports discussed biventricular repair in various other complex cardiac anatomy, and the benefits of 3D cardiac modeling in visualizing complex cardiac anatomy which helped simplify decisions related to surgical options. Riesenkampff et al. produced 11 cardiac models of patients with complex anatomy to investigate the possibility of biventricular repair. There was a unanimous agreement among physicians that these models improved their understanding of the pathology, and helped streamline discussions regarding surgical options for the patients [19]. Valverde et al. described biventricular repair in a 1.5-year-old male with transposition of the great arteries, subaortic VSD, and severe pulmonary stenosis. His 3D cardiac model allowed surgeons to make a decision between the Nikaidoh procedure or the Rastelli repair, and helped them anticipate procedural complications [31]. A case reported by Farooqi et al. involved a seven-year-old male with double outlet right ventricle (DORV) who had previously undergone a bidirectional Glenn procedure. A cardiac model allowed them to visualize the surgical approach and identify a possible baffle position for VSD repair [20]. Garekar et al. printed five cardiac models with different anatomy to study potential surgical options in each case. Three of the five cases underwent successful biventricular repair and surgeons reported a high level of accuracy between the cardiac model and the patients’ anatomy as observed in the operating theatre [21]. Kappanayil et al. also discussed five cases of complex cardiac anatomy, four of which were previously refused surgery due to high mortality and unpredictable outcomes. Of the five cases, three underwent biventricular repairs after using their cardiac models for anatomical understanding and pre-surgical planning [22]. Anwar et al. used a cardiac model to study relationships of anatomical structures, identify a potential baffle pathway, and for pre-surgical planning in an 8-month-old male with situs inversus, dextrocardia, DORV, and L-malposed great arteries [23]. Bhatla et al. described two separate cases of biventricular repair after viewing each patient’s specific cardiac anatomy on a cardiac model. For both cases, the models helped define the spatial relationship between the anatomical structures. They also assisted surgeons in the evaluation of different surgical options and helped them arrive at the most appropriate surgical approach [24,32]. Hoashi et al. printed 20 cardiac models of patients with congenital heart diseases to assess its utility in preoperative surgical simulation. The team found 3D cardiac models very helpful in educating and preparing young surgeons with no prior experience for biventricular repair or neonatal open-heart surgery [25]. Vettukattil et al. described cardiac models for five cases, three of which underwent biventricular repair. These models made the complex anatomy easier to understand, and demonstrated the anatomy precisely which helped in identifying a surgical plan [26].

As evident from the table and previous reported cases, 3D-printed models have helped in understanding the anatomy, spatial relationships of important and adjoining structures, deciding between surgical options, as well as surgical simulations and for educational purposes.

Despite the benefits of 3D printing, the high costs involved in the initial set-up can prevent widespread use of the technology [22,33,34,35]. Use of 3D printing is also limited due to the long processing time needed to print a model. This is dependent on the type of printer, availability of materials, and complexity of pathology to be printed [33,34]. In terms of technical limitations, 3D printed cardiac models do not showcase the dynamic changes that occur during a cardiac cycle which is essential for the understanding of pathophysiology in congenital heart diseases [28,36]. There can also be instances where internal structures like atrioventricular valves, semilunar valves or papillary muscles are not appropriately captured due to blood pooling and the limitations of the imaging technology, resulting in models that are difficult to analyze [30,37]. Lastly, materials currently used to produce 3D cardiac models are unable to replicate tissue texture and properties of the layers of the heart, hence limiting the experience of surgical simulation [30,34]. 

However, its ability to provide tactile experience, valuable information for pre-surgical planning, and improve patient communication proves that it has the potential to be a useful tool in the future of congenital heart surgery [22]. 

## 4. Conclusions

This case highlights the use of a 3D-printed model in helping plan a unique surgical approach to achieve a successful biventricular repair in a patient with heterotaxy syndrome. 3D printing is proving to be an integral tool in understanding the anatomical complexity, planning the surgical approach, and achieving successful outcomes for patients with complex congenital heart defects.

## Figures and Tables

**Figure 1 children-09-00184-f001:**
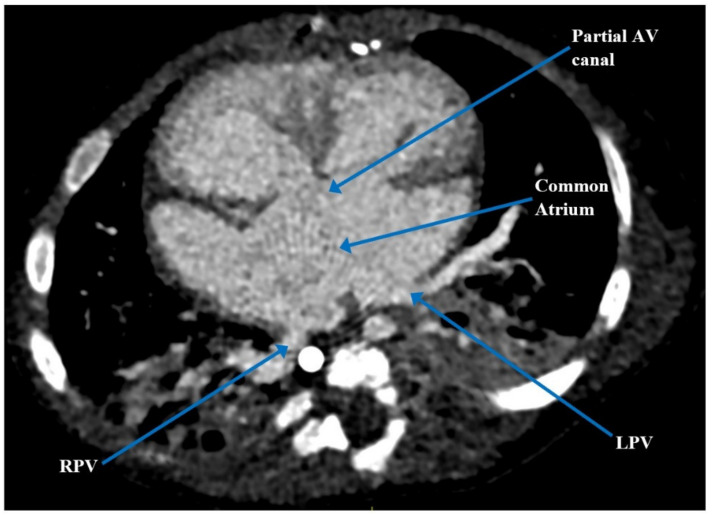
Pre-operative computerized tomography demonstrating atrial situs ambiguous with a common atrium, partial AV canal and ipsilateral pulmonary venous drainage. AV = atrioventricular; LPV = left pulmonary vein; RPV = right pulmonary vein.

**Figure 2 children-09-00184-f002:**
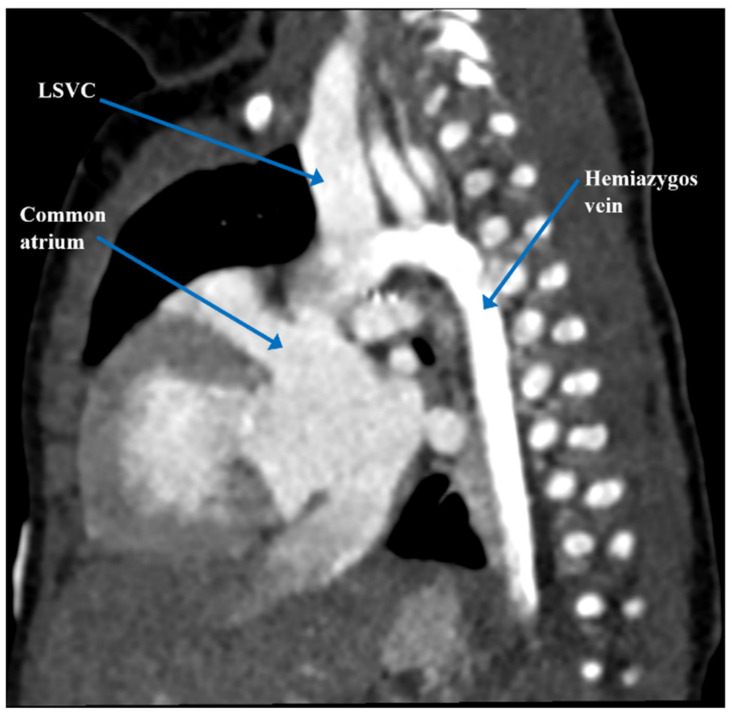
Pre-operative computerized tomography demonstrating hemiazygos vein draining into left SVC and subsequently into the left and anterior aspect of common atrium. LSVC = left superior vena cava.

**Figure 3 children-09-00184-f003:**
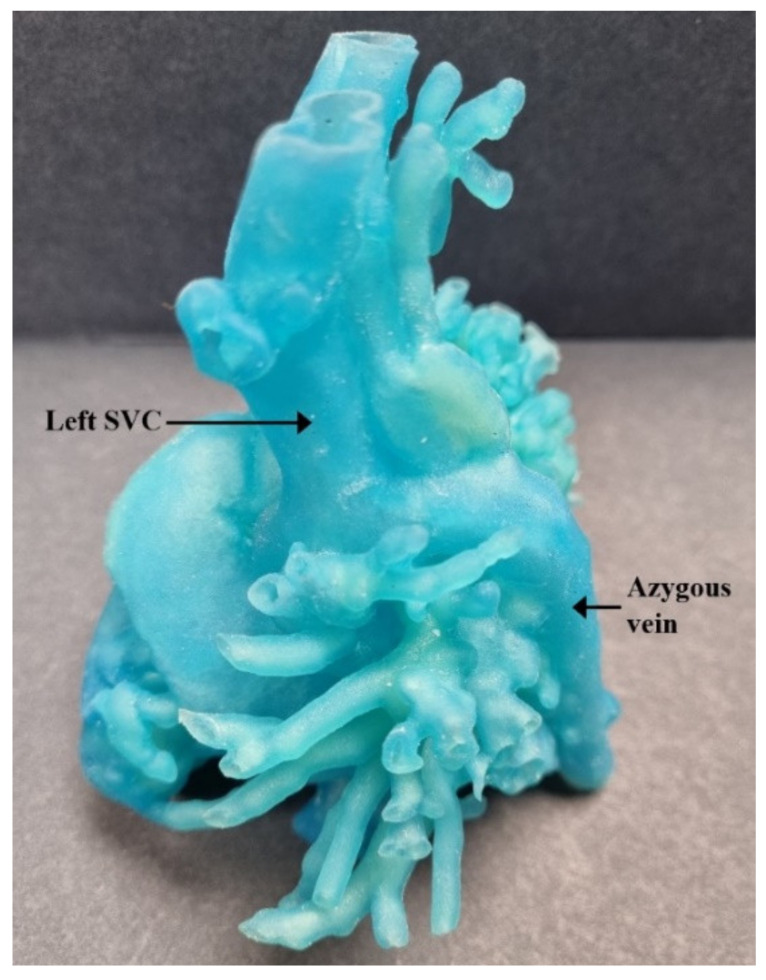
Pre-operative 3D cardiac model demonstrating relationship between hemiazygos vein and left SVC. SVC = superior vena cava.

**Figure 4 children-09-00184-f004:**
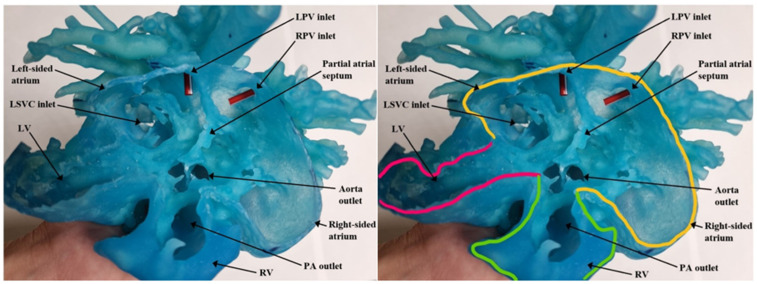
Apical view of the pre-operative 3D cardiac model with common atrium cut open demonstrating ipsilateral pulmonary veins and partial AV canal. Picture on the right shows the outline of the heart chambers. LPV = left pulmonary vein; LSVC = left superior vena cava; LV = left ventricle; PA = pulmonary artery; RPV = right pulmonary vein; RV = right ventricle.

**Figure 5 children-09-00184-f005:**
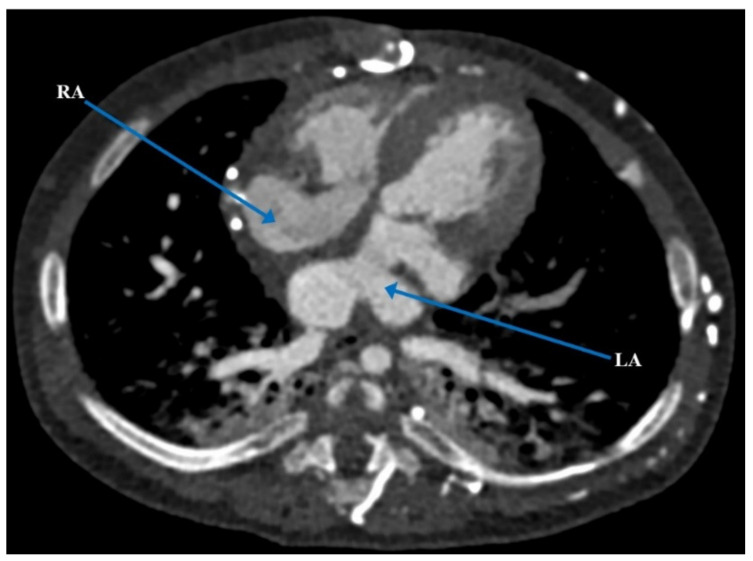
Post-operative computerized tomography demonstrating septation between the left and right atrium. LA = left atrium; RA = right atrium.

**Figure 6 children-09-00184-f006:**
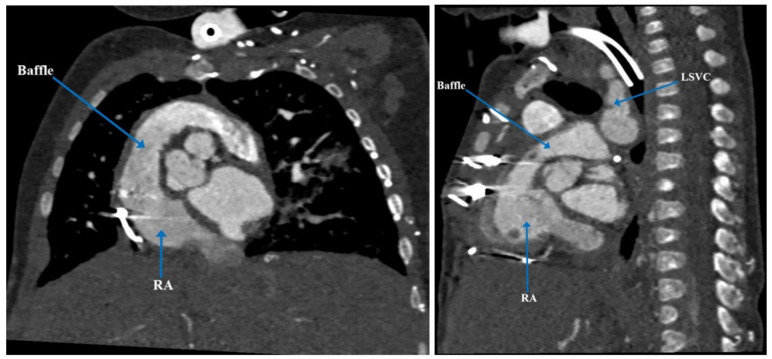
Post-operative computerized tomography coronal and sagittal views demonstrating baffle connection to the right atrium. LSVC = left superior vena cava; RA = right atrium.

**Figure 7 children-09-00184-f007:**
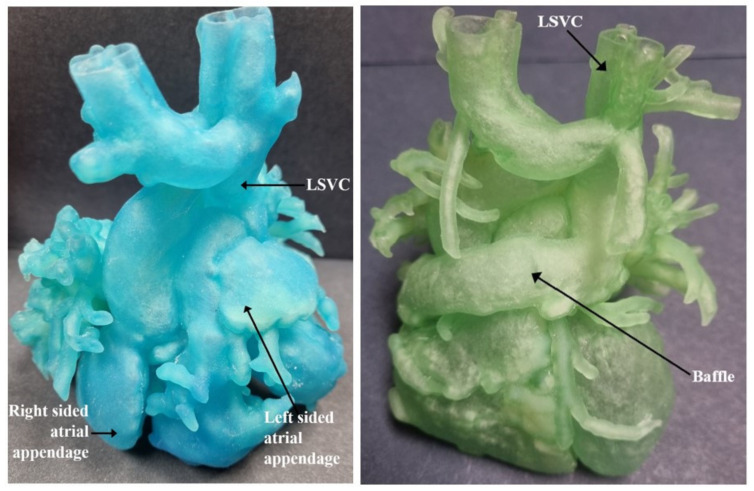
Comparison of pre-operative 3D cardiac model and post-operative 3D cardiac model to demonstrate baffle connection. LSVC = left superior vena cava.

**Table 1 children-09-00184-t001:** Biventricular repair using 3D-printed cardiac models for complex cardiac anatomy.

Authors	CHD Diagnosis	Age	Prior Surgical Intervention	Impact of 3D Printed Model	Surgical Intervention Performed
Riesenkampff et al. [19]	Case 1: DORV, LVOTO, coarctation of the aorta	20 months	PA banding, CoA repair	Defined VSD morphology and its relation to outflow tracts	Biventricular repair with creation of baffle between LV to aorta
	Case 2: DORV, multiple VSDs, valvar pulmonary stenosis	11 months	None		Biventricular repair with creation of unobstructed tunnel from LV to aorta
	Case 3: Tetralogy of Fallot, large VSD, severe pulmonary stenosis, tricuspid valve straddling	9 months	Infundibulectomy		Biventricular repair
	Case 4: L-TGA, pulmonary atresia, VSD	19 years	Enlargement of PA bifurcation		Biventricular repair
Valverde et al. [31]	TGA, subaortic VSD, severe pulmonary stenosis	1.5 years	BT shunt	Defined cardiac anatomy and surgical plan with placement of intraventricular patch	Biventricular repair with Nikaidoh operation
Farooqi et al. [20]	DORV, non-committed VSD	7 years	Bidirectional Glenn procedure	Defined positioning of the ventricles, VSD morphology, and great arteries	Biventricular repair with VSD closure and baffle creation to the aortic root, RV to RPA conduit placement, Glenn anastomosis takedown
Garekar et al. [21]	Case 1: DORV, VSD, side by side great arteries with aorta to the right, mild pulmonary stenosis	7 months	None	Defined cardiac anatomy and helped with surgical approach	Biventricular repair with baffle from LV to aorta
	Case 2: DORV, large VSD, D-TGA, severe pulmonary stenosis	6 years	Initial single ventricle palliation	Defined cardiac anatomy and helped decide single ventricle vs. biventricular repair	Biventricular repair with LV to aorta baffle and RV to PA conduit
	Case 3: DORV, L-TGA, large VSD, severe pulmonary stenosis	11 years	Bidirectional Glenn procedure	Defined cardiac anatomy and pre-surgical plan (Hemi-mustard procedure)	1.5 ventricular repair with hemi-mustard procedure (Bidirectional Glenn left intact)
Kappanayil et al. [22]	Case 1: Situs inversus, dextrocardia, DORV, large VSD, side by side great arteries, severe pulmonary stenosis, bilateral SVC	18 years	None	Defined VSD morphology and its relation to outflow tracts, and challenges in routing LV to aorta	Biventricular repair with Rastelli operation
	Case 2: Situs solitus, mesocardia, large conoventricular VSD, mid-muscular and apical VSD, pulmonary atresia, severe tricuspid regurgitation, moderate ventricular dysfunction	12 years	Right modified BT shunt-1 month; Left modified BT shunt-2 years; Bidirectional Glenn shunt-7 years	Defined complex ventricular septum anatomy and surgical plan for intraventricular tunneling	Biventricular repair with Rastelli operation and tricuspid valve repair
	Case 3: Situs inversus, mesocardia, L-TGA, pulmonary atresia, large VSD, moderate RV hypoplasia	15 years	None	Defined complex anatomy and determined inability to perform full biventricular repair	1.5 ventricular repair with atrial switch, Rastelli operation; bidirectional Glenn shunt left intact
Anwar et al. [23]	Situs inversus, dextrocardia, DORV, L-TGA, VSD	8 months	None	Defined VSD morphology and its relation to outflow tracts and surgical approach	Biventricular repair
Valverde et al. [28]	Case 1: DORV, D-TGA, aortic arch hypoplasia	1 month	Hybrid Norwood Procedure	Changed conservative management to biventricular repair	Biventricular repair with arterial switch, VSD closure, aortic arch repair, takedown of PA band
	Case 2: Heterotaxy syndrome, dextrocardia, isomerism of left-sided atrial appendages, bilateral SVCs, interrupted IVC with azygos continuation	34 years	Septation of common atrium with two patch technique at 2 years and 34 years		Biventricular repair with atrial septation, redirection of systemic flow to tricuspid valve and pulmonary venous flow to mitral valve
	7 patients with various diagnoses: DORV, D-TGA, VSD, pulmonary stenosis, coarctation of the aorta, interrupted aortic arch	7 months –1 year	Various interventions: PA banding, BT shunt, atrial septostomy, PDA stenting, intraventricular baffle from LV to aorta, Yasui operation	Maintained plan for biventricular repair	Biventricular repair
	4 patients with various diagnoses: DORV, AV canal defect, VSD, pulmonary stenosis, common atrium, single left SVC	18 months –4 years	Various interventions: PA banding, Bidirectional Glenn shunt	Changed plan from staged single ventricle palliation to biventricular repair	Biventricular repair
	7 patients with various diagnoses: DORV, TGA, VSD, hypoplastic aortic arch, coarctation of the aorta, pulmonary stenosis	1 month to 11 years	Various interventions: PA banding, Bidirectional Glenn shunt, arterial switch operation, coarctation of the aorta repair, LV to PA homograft, VSD closure	Modified plan for biventricular repair	Biventricular repair
Bhatla et al. [32]	DORV, large VSD, hypoplastic aortic arch	2 months	None	Defined relationship of great arteries to VSD and conal septum	Biventricular repair with Yasui operation and aortic arch reconstruction
Bhatla et al. [24]	DORV, aorta rightward and posterior of PA, large oblong VSD nearly intersected with large conal septum	Newborn	None	Defined complex anatomy and surgical plan for VSD closure via TV approach	Biventricular repair with VSD closure via TV approach, LV to aorta baffle, and arterial switch
Hoashi et al. [25]	11 patients with various diagnoses: DORV, L-TGA, D-TGA, interrupted aortic arch type B, tetralogy of Fallot with pulmonary atresia and MAPCAs	1 month–5.9 years	Various interventions: PA banding, BT shunt, hybrid stage 1 palliation with PA banding and PDA stenting, RV-PA conduit with PA-plasty	Defined VSD morphology and its relation to outflow tracts and surgical approaches for inexperienced surgeon	Biventricular repair
Agarwal A. [27]	Heterotaxy syndrome with interrupted IVC with azygos continuation to SVC, atrial situs ambiguous, AV discordance, balanced AV canal defect, L-looped ventricles, VA concordance	4 years	None	Defined complex cardiac anatomy and pre-surgical plan	Biventricular repair with repair of AV canal defect and atrial switch procedure (Mustard-like operation)
Vettukattil et al. [26]	Case 1: Hypoplastic RV, situs inversus totalis, right atrial isomerism, unbalanced AV canal defect, bilateral SVCs, TAPVC	3 years	BT shunt, bidirectional Glenn shunt, ligation of left SVC, TAPVC repair	Defined feasibility of 1.5 ventricular repair and pre-surgical plan	1.5 ventricular repair
	Case 2: DORV with pulmonary atresia, situs inversus totalis, subaortic VSD with single coronary artery, bilateral SVC	8 years	Unknown initial stage palliation, Bidirectional Glenn shunt	Assessed RV for biventricular repair, planned surgical approach for RV to PA conduit	Biventricular repair
	Case 3: Pulmonary atresia with intact ventricular septum	26 years	BT shunt, bidirectional Glenn shunt	Planned right ventriculotomy and defined tricuspid valve anatomy	1.5 ventricular repair
Present case [6]	Heterotaxy syndrome with partial LPA sling, partial AV canal with a common atrium, atrial situs ambiguous, interrupted IVC, VSD, and PDA	5 months	PDA occlusion, clipping and division of the anomalous LPA	Defined complex anatomy and surgical plan for biventricular repair	Biventricular repair

3D = Three-dimensional; AV = Atrioventricular; BT = Blalock–Taussig; CHD = Congenital heart disease; DORV = Double outlet right ventricle; D-TGA = Dextro-transposition of the great arteries; LVOTO = Left ventricular outflow tract obstruction; IVC = Inferior vena cava; LV = Left ventricle; L-TGA = Levo-transposition of the great arteries; LPA = left pulmonary artery; MAPCAs = Major aortopulmonary collateral arteries; PA = Pulmonary artery; PDA = Patent ductus arteriosus; RPA = Right pulmonary artery; RV = Right ventricle; SVC = Superior vena cava; TAPVC = Total anomalous pulmonary venous connection; TV = Tricuspid valve; VA = Ventriculo-arterial; VSD = Ventricular septal defect.

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
