# Peer review of "Utility of Three-Dimensional Printed Model in Biventricular Repair of Complex Congenital Cardiac Defects: Case Report and Review of Literature"

_children, 2022, doi:10.3390/children9020184_

Round 1

Reviewer 1 Report

The authors present a single case of using a 3D printed model to help plan a complex intra-atrial baffle repair of an infant with heterotaxy syndrome. The pictures are quite good. It does feel like there is an excessive amount of discussion for a single case report and I think the manuscript can be trimmed quite a bit.

Specific points:

Figure 2 and the text are inconsistent. Is it a hemi-azygous vein to the left SVC (which is should be), or an azygous vein (as labelled in Figure 2 and the legend)?

The detailed discussion about the 3D printing process is unnecessary at this point. Anyone interested in reading this article should already have some familiarity. You can simply say that a flexible 3D printed was created using data from the CT scan (to practice operating on?).

The detailed surgical information is likely beyond the scope of this journal (I would suspect mostly a non-surgical audience). You could simply say that a complex intra-atrial baffle was created to route the systemic venous return to the right atrium.

The Table and extremely detailed discussion about prior reports is unnecessary. You can simply state that there have been numerous prior reports of using 3D printed models to help in surgical planning (and cite the prior work). Then state that your case is unique as it appears to be the first description of a modified Warden procedure. Although anecdotally, I had a patient with essentially the same anatomy recently undergo very similar intra-atrial baffling, so be careful with how strongly you emphasize the novelty of the procedure and perhaps focus on the novelty of the use of the 3D model to help.

Reviewer 2 Report

I read author's manuscript describing use of 3D printing for repair of anomalous systemic venous connection with great interest. Here are my comments:

  1. What does the phrase "ipsilateral pulmonary veins" mean?
  2. Do the authors feel that without 3D printing, the baffle was impossible to perform?
  3. There are softwares available which can be used in a virtual reality format to actually go inside the heart and look at the defect. Using certain softwares, looking at the computer screen, patches can be placed, removed and assessed for adequacy and if not adequate, the software model can be reset and virtual attempts can be made again without investing in printers and materials. In light of this new technology, what do the authors think about utility of a hard printed cardiac model?

Thank you.

Round 2

Reviewer 2 Report

I thank the authors for their response. I would like them to include a summary of their responses to my questions in 2-3 lines to highlight the benefits of the 3D printing technology to make it more beneficial to the general populus.

This manuscript is a resubmission of an earlier submission. The following is a list of the peer review reports and author responses from that submission.

Round 1

Reviewer 1 Report

To the Authors

I read with interest the article by Betancourt and colleagues “ Utility of Three-Dimensional printed model in biventricular repair of complex congenital cardiac defects: Case report and review of literature

Among the remarks and commentary that I would submit,

This is a well-written manuscript which relates to the emerging imaging modality of 3-D printing in CHD.

Unfortunately, this case report by itself reveals the current limitation of the technology :

As Angio-CT and angio-MRI studies (with 3D post-processing images), the 3D-printing technology is efficient to evaluate/illustrate extra-cardiac structures mostly.

The intracardiac chambers resolution of current technology is minimal, and mostly used for the evaluation of intra-cardiac remote VSD and the alternative surgical options (tunneling, Yasui, Nikaidoh, uniV pathway).

The relevance of this manuscript is weak, as many surgeons would have consider the left SVC transfer to the right atrial appendage based on the published experience as well as on the available 3D angio-CT. The intracardiac anomalies in the case report is the one of a simple partial AVSD, for which nothing more than a good transthoracic and perioperative transoesophageal echocardiogram is needed.

Secondly, the repeats from the table and the text discussion are numerous, and the most of the review dataset is made of case report of single case, so that I fear there is little to gain from reading this review.

As pointed out several times, the current best indication for this emerging technology is « knowledge translational transfer for all those who have not gained enough expertise in transforming the current echographic, Angio-CT and MRI images into a 3-D accurate estimate of the congenital heart malformation.

I do not really see how to significantly improve the present article as a review, considering it as another case report on the topic might be an option.

Reviewer 2 Report

Summary: 

The authors present a case of heterotaxy syndrome and how they used a 3D printed cardiac model to plan an extracardiac repair of an anomalous drainage of a left superior vena cava to the left-sided atrium, and an intracardiac repair of anomalous pulmonary venous drainage to ipsilateral atria. In addition, they performed a literature review looking for other reports of using 3D printed cardiac models prior to biventricular repair in complex congenital heart disease. The main contribution of the paper could be the seemingly novel surgical technique used for extracardiac systemic venous reconstruction. 

Major issues: 

  1. The surgical technique used for the extracardiac rerouting of the left SVC, named “a modified Warden procedure” by the authors, is claimed to be a novel technique. This claim would be strengthened by an illustration of the technique, so that it could be contrasted with other published extracardiac techniques (https://doi.org/10.1055/s-0035-1564931https://doi.org/10.1016/S0003-4975(97)00456-6https://doi.org/10.1053/j.pcsu.2008.01.007https://doi.org/10.1053/j.semtcvs.2019.04.009). 
  2. The added value of the 3D printed model over the CT images in planning the surgical repair is not well described by the authors, and its utility is therefore not convincing enough. The paper would greatly benefit from an explanation by the authors as to why the CT images were not sufficient to plan the repair, and why only the 3D printed model allowed for the surgical technique to be chosen (was an intracardiac baffle contemplated first and the 3D model then showed that the pulmonary venous drainage precluded this approach, for example?).
  3. The focus of the literature review is not very pertinent to the case report, as it deals mostly with the utility of 3D printed models in intraventricular repairs and abnormal ventriculoarterial connections, neither of which was an issue in the case the authors report. In addition, better and more extensive reviews have already been published (http://dx.doi.org/10.1136/heartjnl-2020-316943https://doi.org/10.1002/jmrs.268https://doi.org/10.21037/jtd.2019.10.38). The paper would make a much bigger contribution to the published literature if the authors did a review focused on surgical techniques +/- utility of 3D printed models in repair of anomalous systemic venous drainage. 

Minor issues: 

  1. Was the morphology of the atrial appendages really indeterminate? The description and the images of the case seem to be consistent with left atrial isomerism.
  2. Was part of the atrial septum present or not? Is it a partial AV canal with a large secundum ASD or a common atrium with a cleft mitral valve?
  3. In Figure 1, the labelling of the pulmonary veins is inverted.
  4. There is no mention if the LPA sling was repaired.
  5. Line 91 – better to use left-sided and right-sided atrial appendage instead of left and right atrial appendage.

Round 2

Reviewer 1 Report

Unfortunately, the authors came short to provide major and relavant modifications to the initial version. 

Reviewer 2 Report

The authors do not address the major issues from the first review adequately. The literature review is not pertinent to their case and the manuscript would be much more appropriate as a simple case report with graphic illustrations of the surgical procedure in addition to the 3D model.